# Family-Based Treatment for Anxiety, Depression, and ADHD for a Parent and Child

**DOI:** 10.3390/ijerph21040504

**Published:** 2024-04-19

**Authors:** Rachel Yoder, Alyssa Michaud, Amanda Feagans, Kendra E. Hinton-Froese, Allison Meyer, Victoria A. Powers, Leah Stalnaker, Melissa K. Hord

**Affiliations:** Department of Psychiatry, Indiana University School of Medicine, Indianapolis, IN 46202, USAfeagansa@iu.edu (A.F.); kendhint@iu.edu (K.E.H.-F.);

**Keywords:** family-based treatment, cognitive behavioral therapy, parent management training

## Abstract

Children with mental illness commonly live with caregivers who suffer from mental illness. Integrated mental-health-treatment approaches can provide more convenient and comprehensive care for families. This case report describes family-based treatment (FBT) for one parent/child dyad. The parent was a 37-year-old female with a history of anxiety and major depressive disorder and concern for symptoms of attention-deficit/hyperactivity disorder (ADHD). The child was an 8-year-old female with generalized anxiety disorder and concern for ADHD and behavioral problems. The parent received individual cognitive behavioral therapy (CBT) and parent management training. The child received CBT. Both also received medication management. The FBT team met regularly for coordinated treatment planning. Self-reported assessments via the Child Behavior Checklist showed meaningful improvement; anxiety decreased to nonclinical range week 12 and depression decreased to nonclinical range week 8. Clinician assessments showed improvement for both patients. Though more time intensive, FBT can yield significant improvement, particularly for children. Pragmatic approaches to treatment planning are important to minimize barriers to FBT.

## 1. Introduction

Mental health problems in children and adolescents; including attention-deficit/hyperactivity disorder (ADHD), disruptive behavior disorders, anxiety, and depression; are common, are frequently comorbid, and can have lifelong adverse effects [1,2]. Evidence-based treatments have generally positive but varying effects, and long-term outcomes of these multifactorial disorders are considered modest, necessitating examination of factors to enhance and individualize treatment [3,4,5,6].

Children with mental illness commonly live with caregivers who also suffer from mental illness, which conveys both a genetic and environmental risk for the development and maintenance of mental health problems in children. When caregiver mental health and family functioning improve, the mental health of the children often improves, even when the children do not receive treatment themselves [7,8]. However, mental health treatment for children is frequently individual, with child-centric therapy and medication management. Although caregivers are often encouraged to obtain their own mental healthcare, barriers; including cost, limited time off from work, and difficulty with appointment scheduling and coordination; make obtaining and maintaining mental healthcare for themselves and their children difficult, if not impossible, for many families [9].

Integrated mental healthcare for children and their caregivers, or family-based treatment (FBT), is a treatment model that can provide more convenient and comprehensive care for families by simultaneously addressing mental health problems for children and their caregivers [10]. Understanding the multiple familial and social factors that may be contributing to mental illness of all family members additionally allows for a more informed, detailed, and individualized treatment approach for both the child and caregiver. Multiple approaches to FBT have been developed [10,11]. This case report describes a referral-based model within a large academic child psychiatry clinic.

Family-based-treatment model description: In coordinated appointments every 1–2 weeks, caregivers and children are provided evidence-based treatment in the same location or via telehealth, often within the same timeframe. The FBT team consists of an adult clinical psychologist, a child-and-adolescent clinical psychologist, and a psychiatrist who is board-certified in both adult and child-and-adolescent psychiatry. Mental health concerns seen within the FBT program include primary concerns of anxiety and depression in caregivers and primary concerns of anxiety, depression, ADHD, and disruptive behavior problems in children. Caregivers and children with diagnoses of anxiety and depression receive cognitive behavioral therapy (CBT). Caregivers with children who have ADHD or disruptive behaviors additionally receive parent management training (PMT). Following initial assessment, therapy initiation, and team and family discussion, the child and/or the caregiver may also receive medication management. At regular intervals, approximately every 1 to 2 weeks, all care providers meet to develop family-based treatment plans that incorporate knowledge of family stressors and prioritization of family needs and goals.

## 2. Detailed Case Description

Two patients, a parent/child dyad, are the focus of this case report. Their case was chosen because they were representative of families typically seen within the FBT program, as related to level of severity, clinical complexity, and comorbid mental health concerns. Further, their consistent involvement in our clinic provided a good illustration of the FBT interventions and coordination of care. They provided written consent to be included. Initial assessment included use of standardized diagnostic assessments: Kiddie Schedule for Affective Disorders and Schizophrenia (child) [12], Mini International Neuropsychiatric Interview 7.0.2-Adult (parent) [13], and self-reported questionnaires.

### 2.1. Parent Intake Information

The parent is a 37-year-old female with a history of anxiety and major depressive disorder (MDD). She suspects that she has ADHD, as growing up she had similar issues in school as her daughter, including trouble paying attention and hyperactivity, but she has not previously received treatment for ADHD.

She reports her mental health concerns control her life and make daily tasks, including parenting, very difficult. Her anxiety has worsened over the past 6 months with intrusive thoughts about her children, most frequently about her daughter’s health. She denies panic attacks, agoraphobia, social anxiety, or obsessions/compulsions. She endorses sometimes feeling so down that getting up off the couch feels impossible, and she is unable to do even small tasks some days. Though she can take care of her children, she finds it hard to be motivated to do anything else. She reports that her energy improved since sertraline and trazodone were initiated approximately five years prior; however, those medications did not help with her anhedonia. In the past, she has experienced passive suicidal ideation—the feeling that if something happened and she died, it would be all right. The last time this occurred was a couple of months before initial assessment. She does not identify any intent, plan, or method for self-harm or suicide; however, she reports having no coping mechanisms or ways of combating these thoughts when feeling depressed. Her refractory depressive symptoms and automatic negative thoughts adversely impact her quality of life and achievement of her parenting goals.

Social history: The parent lives with her husband and two children, aged 2 and 8. She currently works as the primary daytime caregiver of her youngest child. She held varying jobs prior to her current role.

Psychiatric history: She has seen several therapists in recent years. Prior therapy approaches were supportive in nature and described as generally unhelpful. Courses of therapy typically did not exceed a few months. She has not had a psychiatric hospitalization. Psychiatric medications are prescribed by her primary-care provider.

Current medications:Sertraline 100 mg. She has been taking this off and on for approximately ten years. She states this is the psychiatric medication that has worked best for her, though she reports periods in which it seems to stop working.Buspirone 15 mg BID.Trazodone 50 mg qhs.

Previous medication trials: Escitalopram caused irritability. Bupropion and citalopram were ineffective. Fluoxetine helped initially, but the positive effect was not sustained.

### 2.2. Child Intake Information

The child patient is an 8-year-old female with a past psychiatric history of generalized anxiety disorder (GAD).

Strengths: Impressive imagination, supportive family, very social and fun.

Parents report significant concerns related to anxiety in a variety of settings (medical, academic, and social), with greatest impairment noted at school. She began falling behind academically in the context of virtual schooling during first grade due to the COVID-19 pandemic. Since that time, the child became very worried about her academic performance and began having symptoms of panic, including hyperventilating and screaming during timed tests. She also refuses to engage in homework.

Parents also report their child seems to have trouble paying attention, often drifts off, has difficulty resuming activities if interrupted, and makes careless mistakes with writing and reading. She often fights with her 2-year-old sibling. She has trouble following instructions and difficulty staying focused to earn a reward. She has significant difficulty organizing things and avoids tasks that require attention. She often forgets important things that she needs to do and where she puts things. She always seems to be fidgeting and moving around a lot. She rarely plays quietly. She interrupts and blurts out at home and frequently talks back to parents. Parents note she tells a lot of tall tales and lies to get out of trouble. Though the intensity and severity of her meltdowns around school have improved moderately, parents identify managing her behaviors at home as a significant challenge. She has never gotten into trouble at school. She denies any suicidal ideation or self-injury. She denies physical or sexual abuse or neglect. She reports some instances of bullying at school, which the family has worked to address.

Past medical/psychiatric/developmental/academic history: She was diagnosed with a chronic gastrointestinal disorder at age 6 with treatment requiring monthly infusions. Normal development. School testing ruled out dyslexia. No previous therapy or psychiatric hospitalizations.

Medications: Fluoxetine 10 mg prescribed by primary-care physician 6 weeks prior to initial clinic visit. The family feels this was beneficial due to decrease in panic symptoms (no longer was hyperventilating or screaming), but she continues to endorse anxious thoughts interfering with academic work.

### 2.3. Family Assessment and Treatment Planning

The parent has had long-standing symptoms of anxiety and depression, with moderate but fluctuating improvement with sertraline. She has had difficulty with sustaining therapy and has trouble identifying and implementing therapeutic skills. She notes symptoms of ADHD, which are more noticeable and distressing in the context of significant demands of parenting. The family experienced major and simultaneous stressors of the child’s chronic-gastrointestinal-disorder diagnosis and home-based virtual learning during her first-grade year, followed by the stressor of the child returning to school. They have significant strengths of clear affection, strong relationships, and high regard for each other, and the parent is motivated for care.

Family therapeutic goals: Reduce contention among family members by targeting parent symptoms of anxiety and depression, increasing parenting skills around child symptoms of anxiety and disruptive behaviors, and increasing child’s coping mechanisms for anxiety. Assess and, if needed, treat potential diagnosis of ADHD for both family members.

Family therapeutic intervention (Table 1): The parent was scheduled with an adult psychologist for CBT and a child psychologist for PMT. PMT appointments later transitioned to CBT in joint parent-and-child appointments. Both parent and child met with the psychiatrist every 4 weeks for medication management.

### 2.4. Measures

The Clinical Global Impression (CGI) is a clinician assessment comprising two companion one-item measures evaluating the following domains: (1) severity of psychopathology (CGI-S) and (2) change from initiation of treatment (CGI-I) [17].

The Child Behavior Checklist (CBCL) is completed by parents and used to detect behavioral and emotional problems in children and adolescents. Scores are normed by age and gender and can be interpreted as falling in the normal, borderline, or clinical behavioral category [18].

The Quick Inventory of Depression (QID) is a self-administered depression assessment. Scores are classified as 0–5 (normal), 6–10 (mild), 11–15 (moderate), 16–20 (severe), and ≥21 (very severe) [19].

The DSM-5 Level 2—Anxiety is a self-reported assessment for generalized anxiety disorder. Scores are classified as 7–15 (none to slight), 16–19 (mild), 20–27 (moderate), and ≥28 (severe) [20].

### 2.5. Outcomes

CGI—Child: Changed from markedly ill at baseline to mildly ill at 12 weeks.

CGI—Parent: Changed from markedly ill at baseline to moderately ill at 12 weeks.

CBCL: Child scored in the clinical range for depression, anxiety, and ADHD at baseline. Depression score decreased to normal range at week 8 and anxiety decreased to normal range at week 12. ADHD scores remained in clinical range at 12 weeks.

QID—Parent self-report: Baseline score was severe (20). This score improved slightly (18), though remained in the “severe” category at 12 weeks.

DSM-5 Level 2—Anxiety—Parent self-report: Baseline score was moderate (27). This score improved to “mild” (16 at 8 weeks), though increased back to moderate level (20 at 12 weeks).

## 3. Discussion

This case illustrates several considerations when approaching mental health treatment for children and their caregivers. Families with caregivers and children who both have mental health concerns are a particularly vulnerable population, as the combination of mental illness and the work of parenting can cause significant strain for caregivers and impair their ability to meet their parenting goals. This, in turn, can lead to feelings of guilt, frustration, and worsening mood and anxiety symptoms. Key to this parent’s treatment course and outcomes were her courageous and thoughtful self-reflection, acknowledgement of her own mental health concerns, and motivation for seeking help.

While the parent and child both showed improvement in anxiety and depression symptoms throughout treatment, the child had more clinically significant improvement. The parent continued to experience severe depression symptoms and moderate anxiety symptoms. The child’s symptoms of depression and anxiety improved relatively quickly, reaching nonclinical levels at weeks 8 and 12, respectively. This improvement was likely related to initiating fluoxetine prior to treatment and seemed to clearly correlate with the parent implementing 1:1 time and praise skills early and consistently. The parent’s lack of significant improvement was in the context of previous refractory depression: side effects with medication changes; ongoing medication adjustments throughout the treatment course; ongoing, though improving, child disruptive behaviors; and additional work of implementing parenting skills throughout treatment.

Careful and ongoing assessment and treatment of comorbid disorders, family capacity to implement therapeutic skills within the context of team-based observations, and treatment planning for both parent and child were essential. For the child, ongoing assessment of ADHD symptoms led to initiation of treatment and a corresponding improvement in behaviors at the 16-week follow-up. The parent unfortunately experienced delays in achieving optimal ADHD pharmacologic treatment, with ongoing frustration and anxiety associated with symptoms of ADHD and difficulty in optimally implementing individual therapeutic skills. During this time, the parent also struggled with implementing more-complex PMT skills, such as the behavioral reward system. FBT team communication and treatment planning were essential for all providers to understand the context of delays in the therapeutic process. Frequent communication allowed all providers to modify treatment as needed, including responsive pharmacologic management, emphasis on building up established therapeutic skills, and flexing approaches to newer skills. Appropriate ADHD treatment for the parent and improvement in the child’s behavioral symptoms correlated with improvement in the parent’s ability to implement skills learned in individual therapy and allowed transition to non-parenting-related concerns (e.g., assertiveness and relationships with adults in her life).

This case illustrates multiple factors that may limit generalizability in varying settings. Though in different locations, the adult and child mental health services share an electronic medical record system, allowing for greater ease of treating both family members. This form of FBT also takes a significant amount of time for families. In sum, the family participated in 45 therapy and medication management appointments over 6 months, and the parent participated in two forms of therapy simultaneously. During this time, the family also attended regular appointments for the child’s chronic gastrointestinal disorder. It is likely that family factors, such as a two-parent household and this parent’s role staying at home, allowed for more flexibility in appointment scheduling. Still, the FBT model required creativity and thoughtfulness in appointment planning, including the primary use of telehealth appointments and linking medication management appointments. The parent also had to utilize strategies around occupying or finding alternative care for her younger child during appointments. For families with barriers to this level of appointment frequency, providers may consider family-based triage and prioritization of therapeutic skills. For example, for families in which the child’s behavioral problems significantly impact all family members, treatment may start with high yield, though relatively less intensive, PMT skills, such as 1:1 time, praise, and effective instruction. Treatment may then transition to individual CBT for the parent(s) to enhance their mood and coping ability and allow more time for relationship development with initial PMT skills, prior to more-intensive PMT interventions, such as a reward system. Family goals, caregiver capacity, and FBT team coordination should guide this appointment sequence.

Other integrated-family mental-health-treatment approaches in multiple settings have shown both child and parent symptom improvement. A small, randomized controlled trial assessed interpersonal therapy (IPT) for mothers of children with both internalizing and externalizing disorders at the same time and same location as their children’s mental health treatment. The IPT included modifications to improve treatment engagement and relationship difficulties that arise in the context of parenting an ill child. The treatment group was compared to children whose mothers were provided referral information. Mothers in the treatment group had significantly greater improvement in depression symptoms at 3 months. At the 9-month follow-up, children with depression whose mothers received IPT had significantly lower levels of depression [21]. A small, randomized controlled trial for adolescents with depression who had parents with depression compared individual therapy for adolescents to a combination of individual therapy and medication management for the adolescent and parent and joint parent/adolescent therapy. Compared to individual treatment, families in the integrated treatment group showed initial improvement in adolescent depression, improvement in parent depression, and feasibility and acceptability for most adolescent patients [22]. Authors noted elements of improvement and benefits of this treatment approach included reinforcing skills with both family members learning the same therapeutic skills, the ability to coordinate skills, i.e., behavioral activation in scheduling activities together, and improved parenting skills around consistency and problem-solving [23]. Results of a larger, randomized controlled trial incorporating family emotional and behavioral assessment, monitoring, and mental health treatment for both children and their caregivers within a pediatric primary-care clinic showed good feasibility and family engagement and greater reductions compared to the treatment-as-usual group in the CBCL emotionally reactive, withdrawn, sleep problem, aggressive behavior, and total-problem scales. Parents reported greater reductions in anxious and depressed symptoms, rule-breaking behavior, internalizing problems, and total problems. The treatment group also reported greater health-related quality of life [10,24].

There are fewer descriptions of patient experiences within integrated-family treatment, an important area for future research. One qualitative assessment of 18 parents following integration of adult-and-child mental health outpatient services showed overall favorable experiences. Parent diagnoses were heterogenous, with the majority experiencing comorbid disorders. All children were less than 6 years old, and their diagnoses included ASD, PTSD, and unspecified neurodevelopmental disorder. Treatment approaches included individual parent therapy and pharmacotherapy, parent/child groups, family therapy, and individual child therapy. The majority of parents noted improvement in both parent and child symptoms and regulation of emotions and behavior, improved quality of parent/child relationship and ability to empathize and attune with the child, and improved parenting skills and family relations and would recommend the treatment to others. Key elements of improvement noted by parents included focus on the family as a whole, flexibility in treatment tailored to the family’s goals and capacity, components of treatment reinforcing each other, practical coordination of treatment, and collaboration between involved professionals. Parents recommended increased coordination, including joint front desk and professionals able to share in review of all medical records, clear understanding of all potential interventions to choose from initially, and a longer time period of phasing out of treatment [25].

There are still fewer descriptions of the child or adolescent’s experience in integrated-family treatment. In an assessment of a treatment approach that included both parent and adolescent individual treatment and joint parent/adolescent therapy sessions, the treatment approach was acceptable to most adolescents, but some adolescents reported they did not like the addition of joint therapy sessions. A minority of adolescents refused to participate in joint therapy sessions and reported they did not want to hear about their parent’s mood or stressors in their parent’s life. Adolescents also expressed concern about confidentiality with greater degree of parent involvement in their treatment [22,23].

While there is growing evidence for therapeutic modalities that incorporate parenting skills training in addition to individual child treatment, few modalities directly target parent mental health [26]. Further research to develop the best approaches to incorporate parent mental healthcare while minimizing treatment burden is warranted. These could include thoughtful mechanisms to sequence and reinforce core and overlapping components of all treatments. Additional research may guide varying approaches based on parent and child diagnoses and age groups and best practices in settings where combined treatments is more difficult (e.g., separate geographic settings, medical record systems, and front offices) to decrease organizational barriers. While current guidelines around confidentiality are certainly relevant, additional considerations and care when considering family approaches for adolescents are essential for this age group.

## 4. Conclusions

For many particularly vulnerable families with caregivers and children with mental illness, individual and fragmented approaches to treatment are often insufficient. A family-based treatment approach, though potentially more time intensive, can yield significant improvement, particularly for children. Pragmatic approaches to scheduling and treatment planning, ongoing assessment, team-based communication, and comprehensive and coordinated approaches to optimally meet family goals are key components of FBT.

## Figures and Tables

**Table 1 ijerph-21-00504-t001:** Treatment course.

Sessions	Parent Psychotherapy	Child Psychotherapy	Medication Management
1–3	Focused on socialization to treatment and the CBT model, building insight into symptoms, and introducing behavioral skills to address depression (behavioral activation) and anxiety (relaxation techniques) [14]. Started behavioral activation.	PMT: Focused on understanding the child patient’s symptom history and current functional issues related to symptoms. Introduced common reasons for child misbehavior. Instructed caregivers on use of one-on-one time with the child and praising appropriate behaviors [15].	Parent: Trialed extended-release methylphenidate 18 mg. Continued trazodone 100 mg qhs, buspirone 15 mg BID, sertraline 100 mg. At follow-up, the parent reported good tolerability and possible early benefits. Child: Increased fluoxetine in 10 mg increments to 30 mg. Family reported symptom improvement. Teacher Vanderbilt scores assessed after 1 month of school were not concerning for ADHD-related symptoms. Continued to monitor for ADHD-related symptoms with caregivers.
Scores	Depression: 20 (Severe) Anxiety: 27 (Moderate)	Anxiety: 79 (Clinical) Depression: 70 (Clinical) ADHD: 80 (Clinical)	
4–6	Continued with behavioral activation. Identified barriers: Poor recall, distractibility, and perfectionism. Trialed strategies to address these barriers. Introduced cognitive strategies for depression [14].	PMT: Problem-solved issues around one-on-one time, limiting screen time. Supported skills in praise, effective instruction, consistency, and approach to lying/reinforce telling the truth.	Parent: Methylphenidate was increased to 36 mg. Continued other medications. Child: Decreased fluoxetine to 20 mg due to concern for behavioral activation at 30 mg. Teacher requested to repeat Vanderbilt assessments after having more time with the patient and observing poor concentration and worsened school performance.
Scores	Depression: 16 (Severe) Anxiety: 20 (Moderate)	Anxiety: 76 (Clinical) Depression: 70 (Clinical) ADHD: 78 (Clinical)	
7–9	Focused on continuing to use cognitive strategies for depression. Continued with behavioral activation, including building motivation and addressing impact of ADHD symptoms. Identified and worked to address her desire for additional support outside of her family.	PMT: Introduced reward system and discussed reinforcement schedule. Discussed approaches to lower engagement in reward system. Supported continued implementation of previously introduced strategies, such as one-on-one time, enthusiastic praise for appropriate behaviors, and actively ignoring non-dangerous, attention-seeking behaviors.	Parent: Increasing methylphenidate led to unwanted, ego-dystonic, and intrusive thoughts of suicide as well as zoning in on one thing for an unnecessary length of time. Methylphenidate was stopped and bupropion XL 150 mg was started for alternative treatment for ADHD and augmentation of depression treatment. Continued other medications. Child: Repeat parent and teacher Vanderbilts were consistent with ADHD inattentive type. Continued fluoxetine 20 mg, and started methylphenidate HCl 10 mg.
Scores	Depression: 16 (Severe) Anxiety: 16 (Mild)	Anxiety: 67 (Borderline) Depression: 63 (Normal) ADHD: 80 (Clinical)	
10–12	Reviewed treatment plan, including progress towards treatment goals and plan for therapy going forward. Made plans to incorporate other therapy modalities due to mixed response to cognitive strategies and to reduce frequency of appointments to limit stress.	PMT/CBT: Problem-solved behavior around use of screens. Worked with both to develop skills in calm-down techniques. Recommended that parents request a 504 given ADHD and GAD diagnoses in addition to her medical diagnoses.	Parent: Parent did not perceive any change in symptoms on bupropion and preferred an alternate stimulant trial. Stopped bupropion and started lisdexamfetamine 10 mg with plans to increase by 10 mg every 2–3 weeks. Continued other medications. Child: Had good tolerance of methylphenidate HCl and slight initial improvements in behavior. Increased to 20 mg to target inattention and hyperactivity. Continued fluoxetine 20 mg.
Scores	Depression: 18 (Severe) Anxiety: 20 (Moderate)	Anxiety: 64 (Normal) Depression: 66-B (Borderline) ADHD: 78 (Clinical)	
13–16	Introduced acceptance and commitment (ACT) model and incorporated ACT strategies, including values, committed action, and cognitive defusion [16]. Continued working on strategies to address ADHD symptoms. Transitioned to individual focused therapy.	PMT/CBT: Stopped reward system due to child’s refusal to comply. Introduced two-choice method to use instead. Caregivers reported that with consistent prompting, the child patient will perform calm-down techniques. Provided psychoeducation about emotional development and validating and labeling emotions. Made joint plans to transition out of therapy due to improvement.	Parent: Had improvement in mood/executive functioning with lisdexamfetamine; however, experienced increasing symptoms of depression. Increased lisdexamfetamine to 40 mg and sertraline to 150 mg in sequence. Symptoms of ADHD improved, but depression remained moderately severe. Cross-titrated sertraline 150 mg to duloxetine 60 mg for refractory depressive symptoms. Experienced initial partial improvement. Dose was increased to 90 mg. Continued other medications. Transitioned to individual medication management. Child: Improved daytime ADHD symptoms, but struggled in evenings with homework assignments and emotional dysregulation. Started afternoon methylphenidate IR 5 mg. Continued other medications. Transitioned to individual medication management.

## Data Availability

The original contributions presented in the study are included in the article, further inquiries can be directed to the corresponding author.

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
