# Peer review of "Family-Based Treatment for Anxiety, Depression, and ADHD for a Parent and Child"

_ijerph, 2024, doi:10.3390/ijerph21040504_

Round 1
Reviewer 1 Report
Comments and Suggestions for Authors
The topic is relevant and of interest: mental health in children and caregivers in family and school contexts. Caregivers and children with mental health problems are a risk and vulnerable population.
This is a case study describing a family-based treatment (FBT) (treatment model that can provide more convenient and comprehensive care to families by simultaneously addressing the mental health problems of children and their caregivers).
The case is described in detail and the selection criteria are set out. Family assessment and treatment planning are also explained: they were representative of families typically served within the FBT program, relative to level of severity, clinical complexity, and comorbid mental health concerns. However, the children's program for affective disorders and schizophrenia is not described. Examples of items from Mini International Neuropsychiatric Interview 7.0.2- Adult (parent) and self-report questionnaires also do not appear.
Evaluation instruments are not justified. The psychometric characteristics of the Child Behavior Checklist (CBCL), Patient Health Questionnaire (PHQ), and the GAD-7, which is a self-report questionnaire for the detection and measurement of severity of generalized anxiety disorder symptoms. It would be convenient to present examples of items from these instruments.
The objectives of the family intervention should appear clearly and explicitly. They do appear separately, objectives for the daughter and objectives for parents.
The therapeutic intervention and ongoing treatment are described in Table (unnumbered)
In the Discussion section, data are not compared with similar cases that may come from the review of the scientific literature. No studies are cited. Is rare.
It is striking that no reference is made to the follow-up of the case. It doesn't seem to be planned either.
The contributions, limitations and future research do not appear explicitly.
Excessive use of acronyms makes reading difficult, for example, “The father was scheduled with an adult psychologist for CBT and a child psychologist for PMT…”, p. 168-169. It is suggested to avoid using acronyms. It would also be convenient to number the Tables to refer to them.
Reviewer 2 Report
Comments and Suggestions for Authors
The case study concerns an important topic among psychologists, psychiatrists, and relevant researchers.
The manuscript has many merits. However, I have some comments and suggestions that may improve the manuscript.
1) The Introduction is well-written and the study's aims are clear.
2) The intakes and the treatment program are well described.
3) I think that there is room to add the validity and reliability of each measure.
4) For The Child Behavior Checklist (CBCL): The authors described the normal range by percentiles, however, they presented the results by T-scores (see Figure 1), how can the readers know what these scores mean? Thus, I recommend presenting the results by percentiles, or, alternately, describing what the T-scores mean in the measures section.
5) For PHQ9 Parent self-report: The authors stated that "Baseline score showed severe depression (24), which improved to moderate depression over 16 weeks (Line 214-215). However, when looking at Figure 2 it seems that the last dot (circle) describes 18 or even 19 weeks. Thus, the authors should provide a clearer figure.
6) I think that the authors should present the results, even partly, in a table.
7) Did the authors check if the improvements they described were significant?
8) What are the limitations of this case study?
Round 2
Reviewer 2 Report
Comments and Suggestions for Authors
The authors addressed my comments.